# Characterization of direct Purkinje cell outputs to the brainstem

**Christopher H Chen[1,2]\*[†], Zhiyi Yao[1,2†], Shuting Wu[1], Wade G Regehr[1]\***

[1]Department of Neurobiology, Harvard Medical School, Boston, United States;
[2]Department of Neural and Behavioral Sciences, The Pennsylvania State University, Hershey, United States

## eLife Assessment

This **important** paper provides an unbiased landscape for the cerebellar cortical outputs to the brainstem nuclei. By conducting anatomical and physiological analyses of the axonal terminals of Purkinje cells, the data provides **convincing** evidence that Purkinje cells innervate brainstem nuclei directly. The results show that in addition to previously known inputs to vestibular and parabrachial nuclei, Purkinje cells synapse onto the pontine central grey nucleus but have little effect on the locus coeruleus and mesencephalic trigeminal neurons.

**\*For correspondence:**
chc5230@psu.edu (CHC);
wade_regehr@hms.harvard.edu (WGR)

[†]These authors contributed equally to this work

**Abstract** Purkinje cells (PCs) primarily project to cerebellar nuclei but also directly innervate the brainstem. Some PC-brainstem projections have been described previously, but most have not been thoroughly characterized. Here, we use a PC-specific cre line to anatomically and electrophysiologically characterize PC projections to the brainstem. PC synapses are surprisingly widespread, with the highest densities found in the vestibular and parabrachial nuclei. However, there are pronounced regional differences in synaptic densities within both the vestibular and parabrachial nuclei. Large optogenetically evoked PC-IPSCs are preferentially observed in subregions with the highest densities of putative PC boutons, suggesting that PCs selectively influence these areas and the behaviors they regulate. Unexpectedly, the pontine central gray and nearby subnuclei also contained a low density of putative PC boutons, and large PC-IPSCs are observed in a small fraction of cells. We combined electrophysiological recordings with immunohistochemistry to assess the molecular identities of two potential PC targets: PC synapses onto mesencephalic trigeminal neurons were not observed even though these cells are in close proximity to PC boutons; PC synapses onto locus coeruleus neurons are exceedingly rare or absent, even though previous studies concluded that PCs are a major input to these neurons. The availability of a highly selective cre line for PCs allowed us to study functional synapses, while avoiding complications that can accompany the use of viral approaches. We conclude that PCs directly innervate numerous brainstem nuclei, and in many nuclei they strongly inhibit a small fraction of cells. This suggests that PCs selectively target cell types with specific behavioral roles in the brainstem.

## Introduction

Delineating the PC output pathways that mediate signals from the cerebellar cortex to the rest of the brain is a vital step in understanding cerebellar function. Although most studies of PC outputs have focused on the prominent connections to cerebellar nuclei (*Kebschull et al., 2024*; *Hull and Regehr, 2022*), PCs also project directly to the brainstem. Extensive PC projections to vestibular nuclei (*Barmack, 2003*; *Barmack et al., 2000*; *Shin et al., 2011*; *Sekirnjak et al., 2003*) play vital roles in vestibular and ocular-related functions (*Lisberger and Pavelko, 1988*; *Lisberger et al., 1994a*;

*Lisberger et al., 1994b*). PCs also project to the prepositus hypoglossal nucleus, and this pathway may play a role in gaze stabilization (*Cullen, 2023*; *Graham et al., 2023*; *Walberg and Dietrichs, 1988*; *Voogd et al., 1996*). In addition, PCs prominently project to the parabrachial nucleus, which is suited to regulate broad-ranging nonmotor and autonomic functions (*Sugihara et al., 2009*; *De Zeeuw et al., 1994*; *Wylie et al., 1994*; *Chen et al., 2023*; *Hashimoto et al., 2018*).

Several studies suggested that PCs also inhibit cells in other brainstem nuclei that could allow the cerebellum to control diverse behaviors. Rabies-based tracing approaches suggested that PCs are the primary inhibitory inputs to the locus coeruleus (LC) (*Schwarz et al., 2015*; *Sun et al., 2020*; *Breton-Provencher and Sur, 2019*), but this is controversial. These studies have been highly influential and raised the possibility that PCs directly modulate LC-norepinephrine release, and regulate stress, attention, motivation, arousal, pain modulation, reward processing, and other behaviors (*Novello et al., 2024*; *Kang et al., 2021*; *Carlson et al., 2021*; *Zeidler et al., 2020*; *Froula et al., 2023*; *Krohn et al., 2023*). Furthermore, it was also suggested that disruption of the PC-LC pathway contributes to autism spectrum disorder and attention-deficit hyperactivity disorder (*Koevoet et al., 2022*), and impairs the ability of the cerebellum to control seizures (*Streng and Krook-Magnuson, 2021*). However, other studies do not see strong evidence for this PC-LC connection. Retrograde tracing studies of projections to the LC using dye tracers observed minimal PC labeling (*Cedarbaum and Aghajanian, 1978*; *Aston-Jones et al., 1986*; *Aston-Jones and Waterhouse, 2016*), and PC axons largely avoid the LC (*Chen et al., 2023*). Thus, there is no consensus on whether PCs strongly inhibit LC neurons. There is also suggestive evidence that the cerebellum influences additional brainstem nuclei: the injection of trans-synaptic viruses into the posterior vermis labeled cells in pontine central gray (PCG) and nearby nuclei including the LC, Barrington's nucleus, dorsal tegmentum, and others (*Chen et al., 2023*). These findings raise the possibility that direct PC-PCG projections could allow the cerebellum to regulate arousal, sleep-state transitions, micturition, and other behaviors (*Xiao et al., 2023*; *Park and Weber, 2020*; *Keller et al., 2018*; *Hou et al., 2016*; *Verstegen et al., 2017*). However, the viruses used in these studies can also retrogradely label cells (*Zingg et al., 2017*; *Zingg et al., 2020*), and it is therefore necessary to assess whether there are functional PC synapses in these nuclei.

In this study, we determine whether PCs make synapses and inhibit cells within various brainstem nuclei by examining whether PC synaptic boutons are present, and if PCs make functional synapses. Using PC/synaptophysin-tdTomato (*Pcp2^cre^Rosa26^Ai34D^*) mice to label putative PC synapses and map their locations, we find that PCs project widely in the brainstem, and that synaptic densities are highly nonuniform within different nuclei and subnuclei. We identify and characterize PC synapses in brainstem nuclei using optogenetics in PC/ChR2-YFP (*Pcp2^cre^Rosa26^Ai32^*) mice. There was good agreement between the density of PC synaptic boutons and the probability of evoking PC-IPSCs in neurons within different regions and subregions. In regions with a low density of PC boutons, PC-IPSCs were typically observed in a small fraction of cells, raising the possibility that PCs selectively target specific types of neurons in these regions. These findings provide an important resource that will allow future studies to clarify the PC targets in various brainstem nuclei and the behaviors controlled by these output pathways.

## Results
### Visualization of PC projections to the brainstem
We visualized PC synapses in the brainstem of PC/synaptophysin-tdTomato mice in which synaptic boutons are intensely labeled, and axons, dendrites, and somata weakly labeled. We imaged synaptophysin-tdTomato in both sagittal (*Figure 1A*) and coronal sections (*Figure 1B*), and aligned the images to the Allen Common Coordinate Framework (CCF) (*Wang et al., 2020*) based on the contours of the tissue. Abbreviations are as listed in *Table 1*. We collapsed the vestibular subnuclei (MV, LAV, etc.) into the VN, and labeled the territories encompassing the dorsomedial pons as simply the PCG (see Methods). Likewise, the rostral/caudal or medial/lateral divisions of other nuclei are not annotated specifically (PGRN, NTS, etc.).

In low magnification images (*Figure 1A and B*, *left images*), labeling is observed in the cerebellar nuclei and brainstem. In sagittal views, brainstem labeling is sparser in the medial sections than in lateral sections. In coronal sections, the brainstem labeling is densest at bregma −5.85 mm, and sparser in anterior and posterior sections. Medium (*Figure 1A and B*, *second column from left*) and

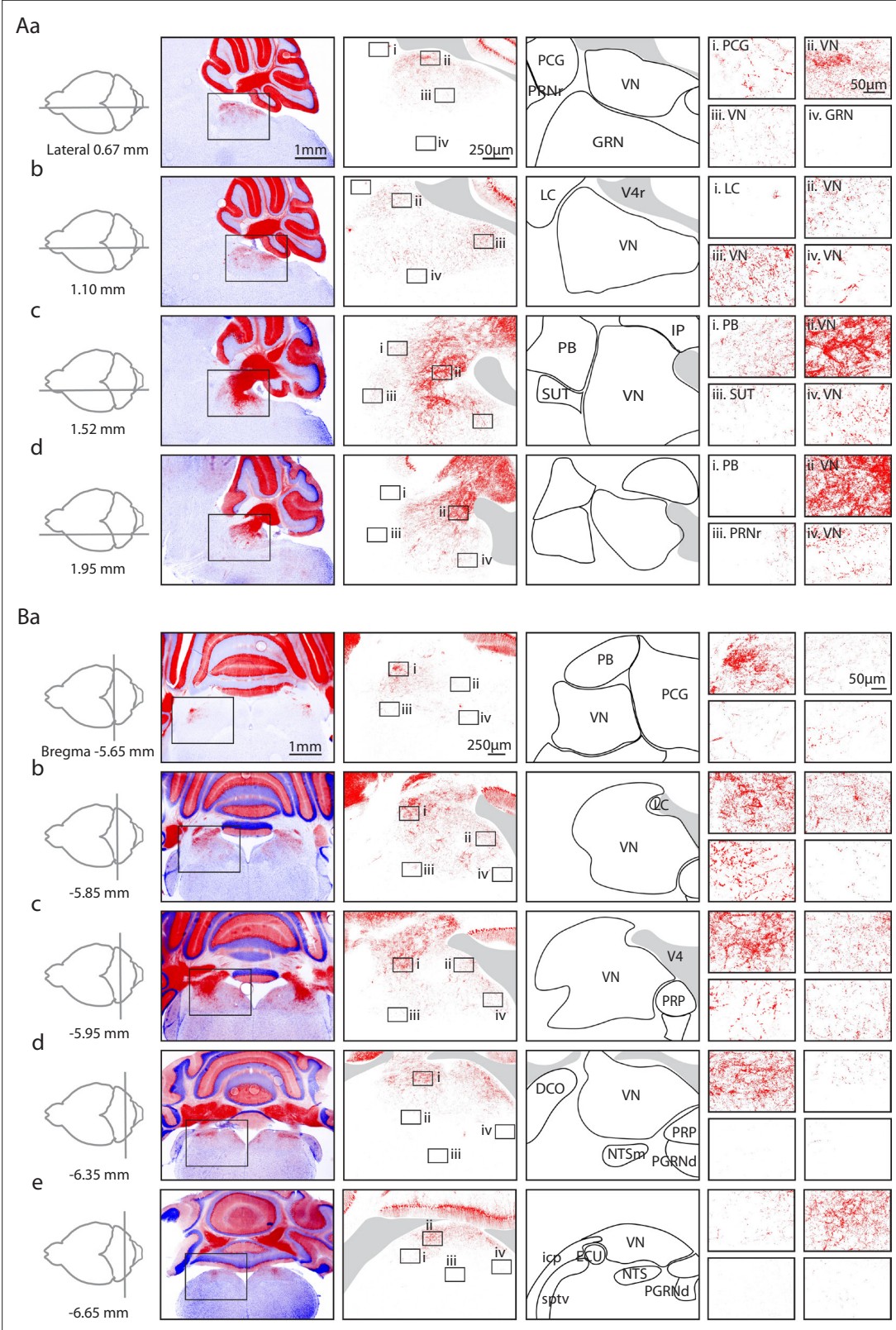

**Figure 1.** Anatomical characterization of Purkinje cell (PC) outputs to the brainstem using PC/synaptophysin-tdTomato mice. (**Aa**) *First column*. Diagram showing parasagittal slice location. *Second column*. Low magnification views of tdTomato (tdT) fluorescence (red) in the cerebellar cortex and brainstem. *Third column*. Medium magnification views of the regions indicated in Ab. *Fourth column*. Anatomical regions corresponding to the preceding column. *Last column*. i–iv. High magnification views of the regions indicated in the preceding column. (**Ab–d**) As in Aa, but for subsequent parasagittal slices. (**B**) As in A, but for coronal slices.

**Table 1.** Abbreviations of brainstem anatomical regions.

| | Nuclei | | |
|---|---|---|---|
| B | Barrington's nucleus | PRP | Prepositus nucleus |
| CU | Cuneate nucleus | SG | Supragenual nucleus |
| DCO | Dorsal cochlear nucleus | SLD | Sublaterodorsal nucleus |
| DTN | Dorsal tegmental nucleus | SPIV | Spinal vestibular nucleus |
| ECU | External cuneate nucleus | SPVI | Spinal nucleus of the trigeminal |
| GRN | Gigantocellular reticular nucleus | SUT | Supratrigeminal nucleus |
| IP | Interposed nucleus | SUV | Superior vestibular nucleus |
| LAV | Lateral vestibular nucleus | VN | Vestibular nucleus |
| LC | Locus coeruleus | X | Nucleus X |
| LDT | Laterodorsal tegmental nucleus | XII | Hypoglossal nucleus |
| MEV | Mesencephalic trigeminal nucleus | Y | Nucleus Y |
| MV | Medial vestibular nucleus | | |
| NI | Nucleus incertus | | **Ventricles** |
| NTS | Nucleus of the solitary tract | V4 | Fourth ventricles |
| NTSl | Nucleus of the solitary tract, lateral | V4r | Fourth ventricle (lateral recess) |
| NTSm | Nucleus of the solitary tract, medial | | **Fiber tracts** |
| PARN | Parvicellular reticular nucleus | arb | Arbor vitae |
| PB | Parabrachial nuclei | icp | Inferior cerebellar peduncle |
| PCG | Pontine central gray | scp | Superior cerebellar peduncle |
| PGRN | Paragigantocellular reticular nucleus | sptv | Spinal tract of the trigeminal nerve |
| PGRNd | Paragigantocellular reticular nucleus, dorsal | VIIn | Facial nerve |
| PRNr | Pontine reticular nucleus | vVIIIn | Vestibular nerve |

high magnification images (*Figure 1A and B i–iv*) showed highly variable densities of PC boutons in different regions. In some regions, labeling was extremely faint for medium magnification images, but putative PC boutons were apparent at higher magnifications (e.g. in the PCG in *Figure 1Aa i and Ba ii*).

There were pronounced regional differences in the extent of labeling within the VN and the PB. In sagittal sections, for the VN the densest PC labeling was observed ventral to the cerebellar nuclei and the fourth ventricle (*Figure 1Ac ii*), and expression fanned out ventrally, anteriorly, and posteriorly. Dense labeling of the VN was also present in the anterior dorsal regions of the other sagittal slices (*Figure 1Aa ii*, *Figure 1Ab ii +iii*, *Figure 1Ac ii*, *Figure 1Ad ii*). Although PC boutons were present throughout the VN, in anterior regions the densities were low near the midline (*Figure 1Aa iii*, *Figure 1Ab iv*) and moderate densities in more lateral sections (*Figure 1Ac iv*, *Figure 1Ad iv*). The density gradient of putative PC boutons within the VN was more apparent in coronal sections. Synaptic densities were highest in dorsal VN (*Figure 1Bb i+ii*, *Figure 1Bc i*, *Figure 1Bd i*, *Figure 1Be ii*), and lower in ventral VN (*Figure 1Ba iii*, *Figure 1Bb iii*, *Figure 1Bc iii*, *Figure 1Bd ii*). A gradient of labeling was also apparent within the PB. The highest densities were present in posterior PB (*Figure 1Ac i and Ba i*). The differences in the densities of putative PC boutons within subregions of the VN and the PB suggest that PCs preferentially regulate specific subregions.

Low levels of labeling were apparent in many other brainstem regions. In the PCG, fluorescence is apparent in high magnification views (*Figure 1Aa i*, *Figure 1Ba ii +iv*). Sparse labeling is present in the PRP (*Figure 1Ba–d iv*), which is consistent with previous studies (*Cullen, 2023*; *Graham et al., 2023*; *Walberg and Dietrichs, 1988*; *Voogd et al., 1996*). Very little labeling was found in the GRN (*Figure 1Aa iv*), the PRNr (*Figure 1Ad iii*), the LC (*Figure 1Ab i*), and the NTS (*Figure 1Be iii*).

These fluorescence images establish that there is considerable variation in the densities of putative PC boutons between and within various brainstem regions. The highest densities of putative PC boutons are present in the range bregma –5.85 mm to –5.95 mm, and 1.5–2 mm from the midline. Notably putative PC boutons were only observed in the dorsal region of the brainstem.

## Quantification of PC synapses in the brainstem

To quantify the densities of PC synapses in the brainstem, we detected individual PC boutons, and registered images to the Allen CCF. Experiments were performed in PC/synaptophysin-tdTomato/CGRP-GFP (*Pcp2^{cre}Rosa26^{Ai34D}Calca^{GFP}*) mice. We used synaptophysin-tdTomato fluorescence to label putative PC boutons, vesicular GABA transporter (vGAT) immunohistochemistry to visualize all GABAergic synapses, and CGRP-GFP to label subnuclei to aid in atlas alignment. The resulting alignment is shown for a coronal slice (*Figure 2A*, see Methods). To quantify synapses, confocal images were taken for a z-stack of 1.5 µm of the dorsal brainstem bilaterally. All regions previously observed with tdTomato signal (*Figure 1*) were imaged, as well as additional sections with no obvious labeling within 100 µm of labeled regions. Since weak tdTomato labeling is observed in the axons of PCs, vGAT immunohistochemistry was used to help identify putative presynaptic sites. In addition, PC contributions to total inhibition in each region can be measured by comparing GABAergic boutons without tdTomato labeling (*Figure 2B*, *lower right, gray dots*) or with tdTomato colabeling (*Figure 2B*, *lower right, red dots*, see Methods). This approach identifies putative presynaptic sites but does not identify functional synapses.

This approach was used to detect putative PC boutons in a series of coronal slices from Bregma –5.45 to –6.95, where the majority of PC projections reside. Inhibitory synapses were present at high densities in all regions in all slices (*Figure 2C*, *left, gray*), while putative PC boutons were spatially restricted (*Figure 2C*, *left, red*). The total number of detected PC boutons, the density of putative PC boutons, and the fraction of inhibitory boutons that are made by PCs were determined for each region (*Figure 2D–F*, data points color-coded to indicate slice position, n=3 mice, 2 hemispheres/mouse). Areas with observable PC boutons were selected for display along with their near neighbors, including example areas with no tdTomato labeling (PGRNd, SPVI, NTSl, NTSm). Areas ventral to displayed regions were not labeled.

High densities of putative PC boutons are present in the superior, lateral, spinal, and medial vestibular nuclei, but there are pronounced regional differences in bouton densities within the subnuclei and related nuclei (*Figure 2C–F*). In the superior vestibular nucleus, putative PC boutons are present at high density and comprise up to 85% of all inhibitory inputs. In the lateral vestibular nucleus, the number, density, and fractional contribution of putative PC boutons were largest in posterior sections. PC inputs contributed to ~40% of all inhibitory inputs in posterior LAV. The medial vestibular nucleus spanned the longest anterior-posteriorly and has the most putative PC boutons in the middle sections, from bregma –6.05 mm to –6.35 mm. There was a pronounced dorsal-ventral gradient in the density of putative PC boutons. The spinal vestibular nucleus is the most posterior nucleus and had less variability across the anterior-posterior axis. It has a high average putative PC bouton number and density across sections, and 26% of inhibition originates from PCs. The prepositus nucleus also receives PC inputs that are highest at bregma –6.05 mm and –6.35 mm and in dorsal regions. PCs contribute up to 9% of the total inhibitory inputs in the prepositus nucleus (*Figure 2D left, middle, right*). In the more posterior sections (bregma –6.65 mm and –6.95 mm), putative PC boutons were restricted to the dorsal region in the vestibular nuclei, the external cuneate, and the cuneate. The external cuneate and the cuneate are small regions, and the densities were correspondingly higher though the absolute number of synapses are relatively low and variable. The external cuneate spanned from bregma –6.65 mm to –6.95 but has more putative PC boutons in the posterior section. It is evident that the vestibular nuclei, the cuneate nucleus, and prepositus nucleus are not uniformly targeted by PCs.

There is a posterior to anterior gradient of putative PC presynaptic boutons in the PB. In anterior sections (*Figure 2C*, bregma –5.45 mm to –5.65 mm) high densities of putative PC boutons were only apparent in a few localized regions of the PB. The medial-ventral region of the PB (bregma –5.65 mm) has extensive PC inputs, comparable in density to some regions of the vestibular nuclei.

PC synapses are present but at low densities within the PCG and nearby subnuclei including the LC, Barrington's nucleus, sublaterodorsal nucleus, laterodorsal tegmental nucleus, dorsal tegmental nucleus, and supragenual nucleus (*Figure 2C*, bregma –5.45 mm to –5.65 mm). Putative PC boutons

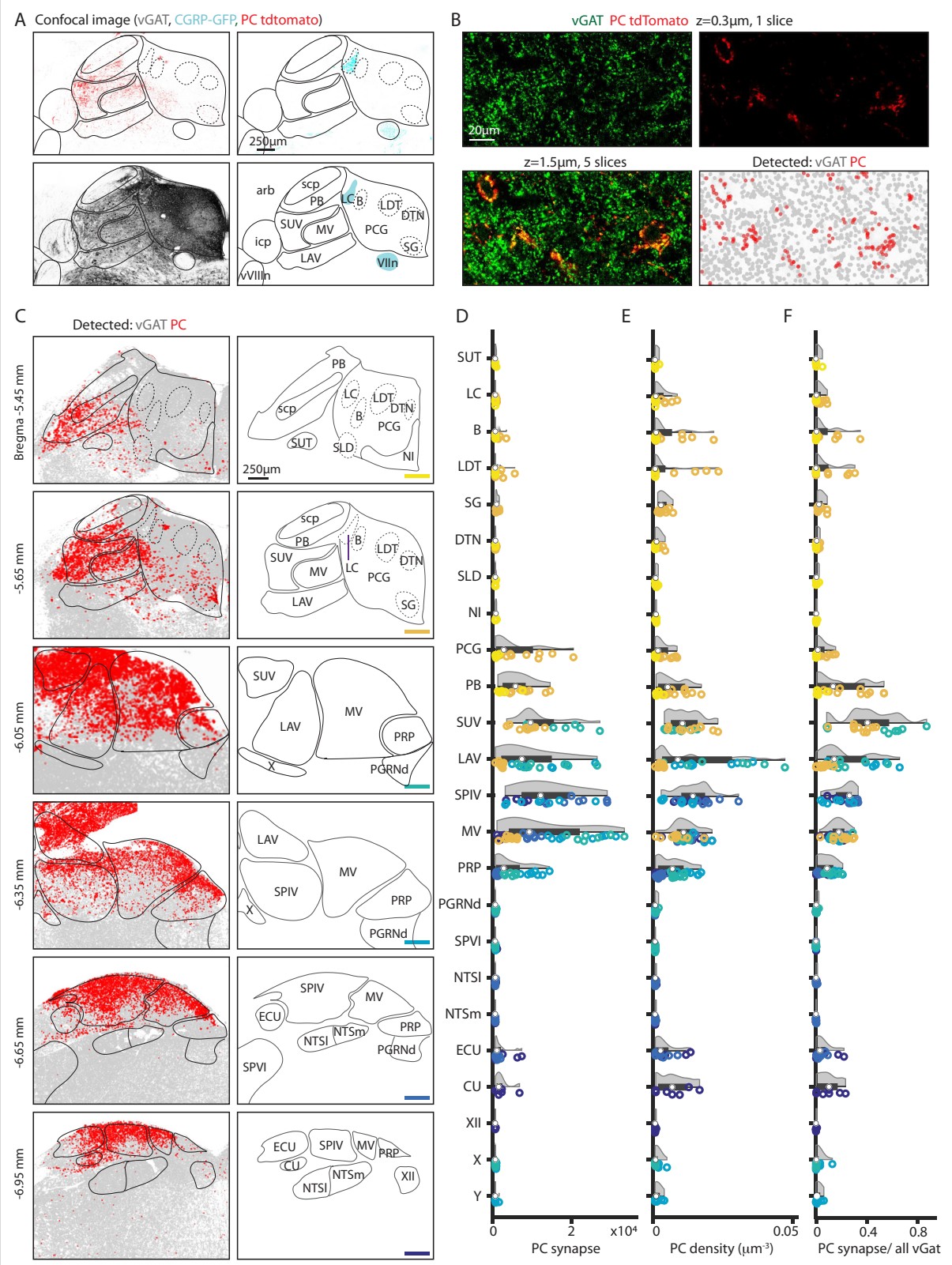

**Figure 2.** Quantification of putative Purkinje cell (PC) boutons in the brainstem. GABAergic boutons were detected using a vesicular GABA transporter (vGAT) antibody and PC presynaptic boutons were detected based on tdTomato-synaptophysin labeling in PC/synaptophysin-tdTomato mice. (**A**) (*Upper left*) Confocal image of td-Tomato fluorescence. (*Upper right*) GFP fluorescence in CGRP-GFP mouse in which the locus coeruleus and facial nerve are labeled. (*Lower left*) vGAT labeling of GABAergic boutons. (*Lower right*) Allen Atlas coronal section corresponding to confocal images with

*Figure 2 continued on next page*

*Figure 2 continued*

landmarks used for registration highlighted in cyan. (**B**) (*Upper left*) vGAT labeling in a single confocal section. (*Upper right*) tdT labeling in a single confocal section. (*Lower left*) z-Stack of vGAT and tdT labeling in five sections (1.5 μm thick). (*Lower right*) Detected GABAergic boutons based on vGAT labeling (*gray*) and PC boutons (*red*). (**C**) (*Left*) Detected GABAergic boutons corresponding to PCs (*red*), and neurons other than PCs (*gray*) from six example coronal sections. (*Right*) Labeled anatomic regions for each coronal section. (**D**) Number of putative PC presynaptic boutons detected per section in each region. Individual section values are represented with symbol colors correspond to the sections in C, right (bottom-right corner color bars). Violin plot of the region shown in gray with average and quartile values in black. (**E**) Density of PC boutons in each region. Individual section and region average as represented in D. (**F**) Percentages of GABAergic boutons that correspond to PCs for each region. Individual section and region average as represented in D.

make up a small fraction of inhibitory synapses in these areas (*Figure 2F*). The Barrington's nucleus and the laterodorsal tegmental nucleus had a higher PC density and fraction of inhibition, although it is important to note that these PCG subregions are relatively small in area. The LC, which is labeled in CGRP mice, had few putative PC boutons with a slightly elevated PC density count in a few posterior sections.

## Characterization of functional properties of PC synapses in the brainstem

To confirm these PC connections in different brainstem regions and examine their strength, we optically stimulated PC fibers in coronal brain slices of the dorsal brainstem in PC/ChR2-YFP (*Pcp2^{cre}-Rosa26^{Ai32}*) mice and measured IPSCs. It is likely that this approach provides a good estimate of connectivity even when PC axons are cut, because severed axons expressing ChR2 can be optogenetically activated to evoke synaptic responses (*Mao et al., 2011*; *Petreanu et al., 2007*; *Petreanu et al., 2009*; *Jackman et al., 2014*). We recorded optically evoked IPSCs in 133 of 310 neurons in the dorsal brainstem (between bregma –5.45 mm and –6.35 mm) to generate a physiological map of PC inputs to the brainstem. The IPSC amplitudes of the recorded brainstem neurons (*Figure 3A*, gray-scale color-coded) are shown along with the density maps of putative PC boutons from *Figure 2*. For the density map, PC bouton counts were binned into $144 \times 144 \times 4 \ \mu m^3$ volumes, and averaged across the multiple sections. The likelihood of optically evoking a PC-IPSC varied with coronal section from anterior to posterior was 9%, 32%, 51%, and 45% (–5.45, –5.65, –6.05, and –6.35 mm to Bregma). We quantified the response amplitudes (*Figure 3B*), probability of evoking a PC-IPSC (*Figure 3C*), and the IPSC amplitudes (*Figure 3D*), and displayed results of notable regions.

In the most anterior slices (Bregma –5.45 mm and –5.65 mm), PC-IPSCs were not evoked at most locations (47/53 cells nonresponding), which is consistent with the very low density of PC synaptic boutons (*Figure 3A*). For the PCG, no PC-IPSCs were evoked in the Bregma –5.45 mm section. In the Bregma –5.65 mm section PC-IPSCs were evoked in 23% of cells, with highly variable amplitudes (*Figure 3B*), and the average PC-IPSC in responding cells was over 1 nA (*Figure 3D*). For the PB, PC-IPSCs were evoked with a low response probability in Bregma –5.45 mm slices, but at Bregma –5.65 mm, 48% of cells responded (*Figure 3C*). In the PB, the amplitudes of IPSCs onto different cells were highly variable (50 pA to over 10 nA) (*Figure 3B*), and the average of responding cells was 2 nA.

In more posterior slices (Bregma –6.05 mm and –6.35 mm), PC-IPSCs were evoked in a high percentage of cells in the SUV, LAV, MV, and PRP (45%, 59%, 58%, and 52% respectively, *Figure 3C*). The SUV and LAV had predominantly large IPSCs in responding cells, with most evoked currents larger than 500 pA and the average PC-IPSC of responding cells was several nanoamps. In the SPIV, MV, and PRP, PC-IPSCs ranged from 50 pA to ~2 nA and average IPSC amplitudes of responding cells were less than 1 nA (*Figure 3B and D*). Overall, PC-IPSCs were evoked in a larger percentage of cells in regions with a high density of PC boutons (*Figure 3E*).

## Assessing PC inputs to the LC and mesencephalic trigeminal neurons

There are many types of molecularly distinct neurons in the brainstem (*Nardone et al., 2024*), which could be differentially targeted by PCs and contribute to the highly heterogeneous amplitudes of evoked PC-IPSCs we observed. We therefore took a targeted approach to record from neurons in the MEV and LC, which are distinct neuron populations with well-characterized markers that are in close proximity to PC boutons and to areas with large PC inputs (the PB and vestibular nuclei, *Figure 4A*). Previous trans-synaptic tracing efforts had also suggested potential PC inputs to these cells, but

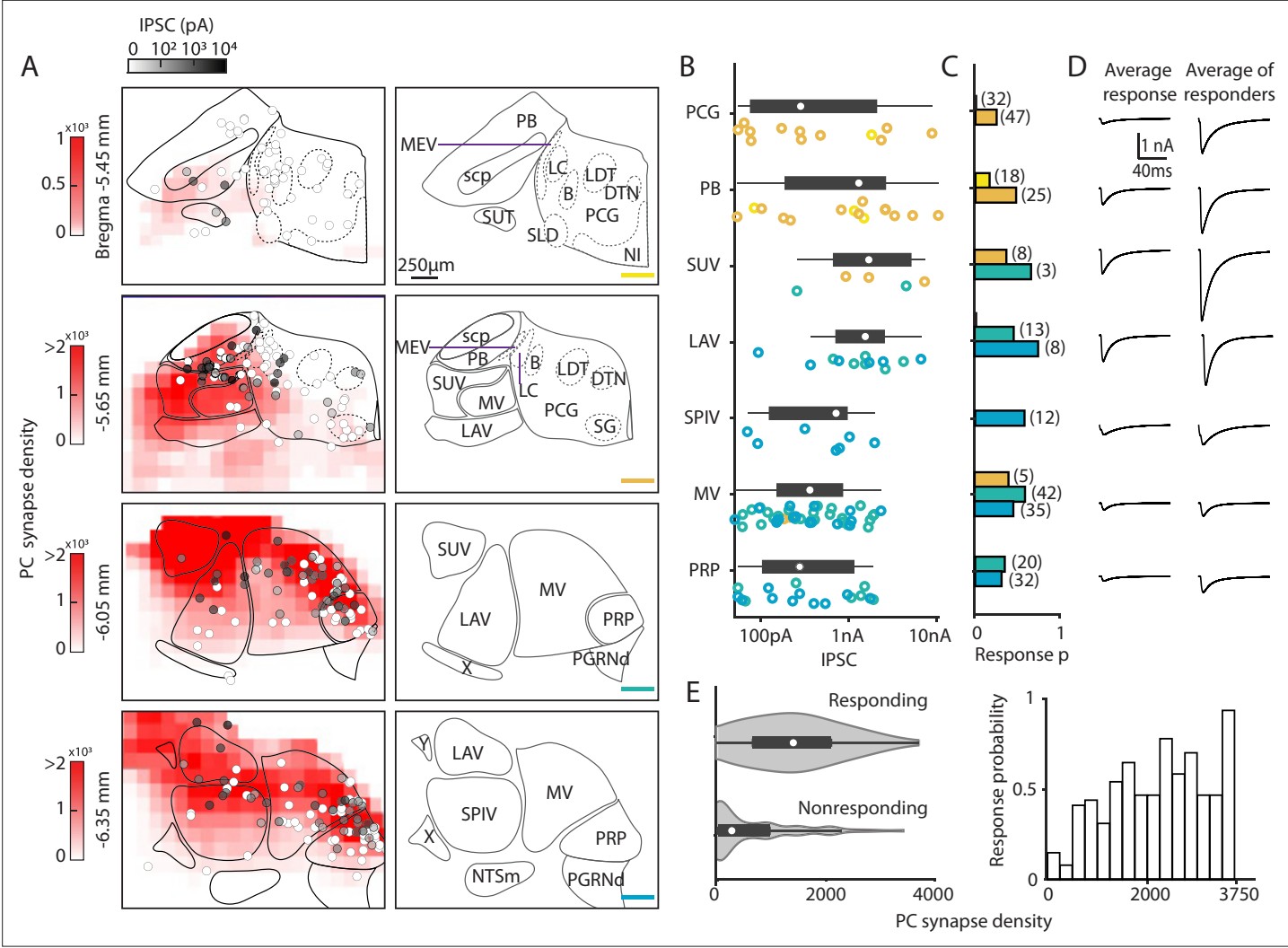

**Figure 3.** Characterization of functional properties of Purkinje cell (PC) synapses in the brainstem. Light-evoked PC-IPSCs were measured in the brainstem of PC/ChR2-YFP mice. (**A**) (*Left*) Averaged putative PC presynaptic bouton density from previous quantification of brainstem regions (red, from *Figure 2*). Synapses were binned in voxels of 144×144×4 μm³. The locations of all recorded neurons are shown with the symbols coded for the light-evoked IPSC amplitudes (n=310). (*Right*) Labeled anatomical regions are indicated and the position of each slice is color-coded, as in *Figure 2*. (**B**) Evoked amplitudes of the responding neurons (n=113) in each region are shown. Symbol colors correspond to the sections in A. (**C**) Fraction of responding cells in each brainstem region and total number of neurons recorded (n=300). Bar graph colors correspond to the sections in A. (**D**) (*Left*) Average current in each region for all cells. (*Right*) Average current of responding cells in each region. (**E**) (*Left*) A violin plot of the average putative PC presynaptic bouton densities for neurons where a light-evoked PC-IPSC was detected (responding) and for cells where such a response was not observed (nonresponding). (*Right*) The probability of observing a light-evoked PC-IPSC in neurons is plotted as a function of the density of putative PC synaptic boutons in that voxel.

functional characterizations of these connections are lacking (*Chen et al., 2023*; *Schwarz et al., 2015*; *Sun et al., 2020*; *Breton-Provencher and Sur, 2019*).

We began by studying the extremely large unipolar neurons of the MEV. These neurons are developmentally part of the periphery, gap junction coupled, exhibit a unique chloride gradient, and directly sense and control the muscles of the jaw (*Curti et al., 2012*; *Florez-Paz et al., 2016*; *Yokomizo et al., 2005*). We used immunohistochemistry for parvalbumin (PV) to identify MEV neurons (*Florez-Paz et al., 2016*). We measured PC-IPSCs, filled cells with biocytin, and identified these neurons based on their shape and PV expression (*Figure 4B*). Recordings were made in both coronal slices (bregma –5.45 mm and –5.65 mm) and horizontal slices, but PC-IPSCs were not evoked in any of the recorded cells (*Figure 4C*, n=23). Thus, despite their close proximity to putative PC boutons, PCs do not provide a significant source of inhibition to MEV neurons.

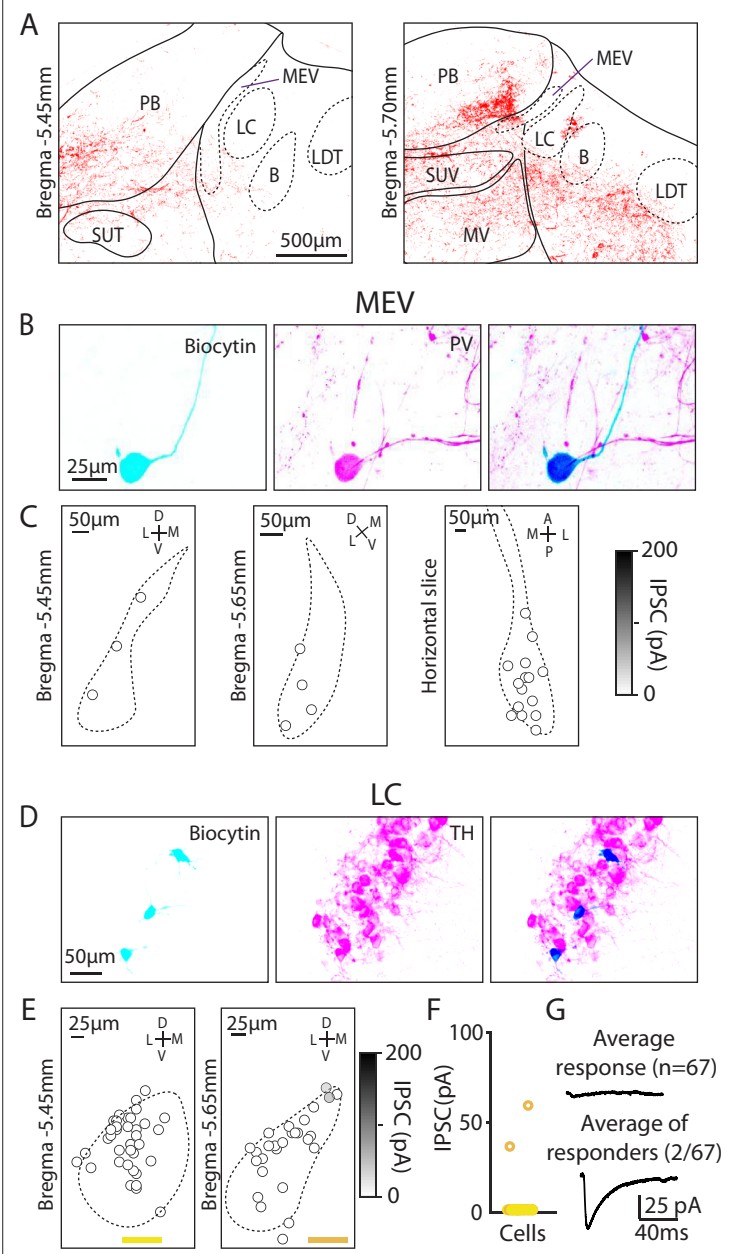

**Figure 4.** Minimal Purkinje cell (PC) inputs to locus coeruleus and mesencephalic trigeminal neurons. (**A**) Example confocal image of PC presynaptic boutons labeled by synaptophysin-tdT showing cases where labeling was near locus coeruleus and the mesencephalic trigeminal nuclei. (**B**) (*Left*) Biocytin was included in the pipette to label MEV cells during whole-cell recordings. (*Middle*) MEV cells visualized with a parvalbumin (PV) antibody. (*Right*) Colocalization of biocytin and PV confirms that patched neurons were PV+MEV neurons. (**C**) Locations of MEV cells in two coronal slices and one horizontal slice. No PC-IPSC responses were detected in any of the cells (n=23) at any location. (**D**) As in B, but for LC cells labeled with a TH antibody. (**E**) As in C, but for two coronal levels of the LC (n=67). (**F**) Summary of the amplitudes of PC-IPSCs recorded in LC cells. (**G**) Average PC-LC neuron IPSCs for all cells (n=65, *left*), and for cells where a synaptic response was detected (n=2, *right*).

We also used optogenetics and immunohistochemistry to examine PC-IPSCs in the LC to directly assess whether PCs provide a major source of inhibition to LC neurons, as has been proposed (*Schwarz et al., 2015*; *Sun et al., 2020*; *Breton-Provencher and Sur, 2019*). The low density of synapses in the LC raised doubts about whether PCs provide a prominent source of inhibition in the LC (*Figure 4A*), but it was possible that PCs might synapse onto LC dendrites that extend beyond the core of the LC.

We therefore directly assessed PC-LC synaptic connectivity by measuring optically evoked PC-IPSCs using a recording electrode containing biocytin and used immunohistochemistry to identify TH-expressing neurons of the LC (*Figure 4B*). We examined the LC at two coronal levels in the posterior LC and found that no PC-IPSCs were evoked in the anterior section (0/27), while small currents were evoked in 2/40 neurons in posterior sections (bregma –5.65 mm) (*Figure 4C and D*). The average PC-IPSC of all cells was 1.4±8.2 pA. Thus, PC-LC synapses are very rare and weak.

## Discussion

### Widespread but nonuniform PC inputs to the brainstem

Here, we find that PCs make direct synapses within numerous brainstem nuclei. The densities of these synapses vary widely in different nuclei. There is also a pronounced gradient of PC synapses across the dorsal-ventral axis of the several brainstem nuclei, indicating that PCs do not uniformly regulate activity within these nuclei. The combination of synaptophysin-TdT expression restricted to PCs and vGAT immunohistochemistry allowed us to quantify the absolute number and density of putative PC boutons, and the fraction of inhibitory inputs that are from PCs for each region. Putative PC boutons comprised up to 75% of the inhibitory synapses in some vestibular regions and 55% in the caudal PB. Thus, PCs make prominent contributions to inhibition in some brainstem regions that are comparable to levels observed in cerebellar nuclei (72%) (*De Zeeuw and Berrebi, 1995*). In contrast, in the PRP and PCG, PCs provide a low total number of inputs (PRP, PCG vs. VN; 0.002, 0.006, >0.01 synapses/$\mu m^2$, respectively) and a small fraction of inhibitory inputs (4%, 8%, 15–50%), where they inhibit only a modest fraction of cells (13%, 33%, 54%). In all of these regions, PCs likely function similarly to their inputs to the cerebellar nuclei, where a very brief pause in firing can lead to large and rapid elevations in target cell firing (*Han et al., 2020*; *Wu et al., 2024*), but PCs regulate the firing of a very different fraction of cells in brainstem nuclei and can do so in a variety of different ways (*Shin et al., 2011*; *Najac and Raman, 2015*; *Özcan et al., 2020*).

### PC synapses in the PCG and nearby regions

We establish that there are PC boutons within the PCG and associated nuclei, and that there are strong PC synapses in this region. This suggests that PCs play a role in regulating activity in these regions and in the behaviors they control. However, the PCG is enigmatic in structure and function, with the most commonly used mouse atlases disagreeing on the borders and identity of subnuclei (*Wang et al., 2020*; *Paxinos and Franklin, 2019*; *Kronman et al., 2023*) (see Methods). Barrington's nucleus, a region that controls micturition (*Hou et al., 2016*; *Verstegen et al., 2017*), stands out as being well defined anatomically and functionally, and is a promising candidate for regulation by PCs. Our previous study raised the possibility that PCs in the posterior vermis might directly target cells in Barrington's nucleus (*Chen et al., 2023*), but this study relied on AAV1 injections in the cerebellum, that in addition to anterogradely labeling PC targets, could have retrogradely labeled cells in Barrington's nucleus if they project to the cerebellar cortex. Our results highlight the challenges in studying PC synapses in the PCG and related nuclei. We observed putative PC boutons and synaptic connections for cells in the vicinity of Barrington's nucleus (*Figures 2 and 3*). As suggestive as these findings are, it remains an open question whether direct PC synapses to Barrington's nucleus control micturition without additional experiments using molecular markers to identify Barrington's nucleus. Similarly, further studies are needed to identify other PC targets in the PCG and determine what behaviors they control.

### Gradients of PC synapses in the brainstem

In general, PCs inputs to brainstem nuclei are spatially heterogenous. The caudal-rostral gradient of putative PC boutons in the PB allows PCs to target specific subtypes of PB neurons (*Chen et al., 2023*). We find that putative PC boutons in the PCG and nearby subnuclei also exhibit a caudal-rostral gradient. Additional studies are needed to identify PC targets in these regions and to determine the significance of their spatial heterogeneity. There is also a pronounced dorsal-ventral gradient in many vestibular subnuclei and related nuclei. Previous studies suggested that PCs do not project to all regions of the VN, but they were unable to quantify the regional densities of putative PC boutons. Labeling individual PCs or those within subregions of the cerebellar cortex helped to determine their

specific projections in the VN (*Barmack, 2003*; *Wylie et al., 1994*; *Shojaku et al., 1987*; *Tabuchi et al., 1989*; *Blot et al., 2023*), but did not label all PC boutons in the VN. Immunohistochemical methods targeting proteins selectively expressed in PCs offer the potential to label all PC boutons in the VN. Calbindin is a widely used PC marker, but its presence in vestibular afferents compromises its use in the VN (*Kevetter, 1996*). PKCγ is also expressed by PCs, but has the advantage that it is not present in vestibular afferents (*Barmack et al., 2000*). The lack of PKCγ immunoreactivity established that PC synapses are absent in the ventral VN, which is qualitatively similar to what we observe. However, the possibility that PKCγ is expressed by other cell types projecting to the VN complicates the use of PKCγ labeling as a definitive marker for identifying PC synapses. Our studies establish that PC synapses are present at high densities throughout the SUV; in the dorsal LAV, MV, PRP, SPIV; in subregions of the ECU; and there are large subregions that are devoid of PC synapses. Multiple lines of evidence suggest that PC synapses in the VN play a role in canceling self-motion related to eye movements and proprioception (*Cullen, 2023*). The distribution of synapses suggests that processing within VN subnuclei is spatially segregated according to the locations of PC synapses, implying that PC feedback potentially cancels signals related to self-motion specifically within these regions.

## Brainstem regulation by direct PC inputs or indirectly via the deep cerebellar nuclei

PCs can also disynaptically influence the brainstem by disinhibiting cells in the deep cerebellar nuclei that in turn project to the brainstem. It is likely that some brainstem regions receive both direct PC input and indirect inputs from the deep cerebellar nuclei (*Novello et al., 2024*; *Fujita et al., 2020*; *Judd et al., 2021*), and that these two different pathways serve different roles. For example, direct and indirect inputs that target different regions of the PB are implicated in different behaviors (*Chen et al., 2023*; *Hwang et al., 2023*). The deep cerebellar nuclei projections to the vestibular nuclei are not well understood, but they are likely a mix of excitatory and inhibitory inputs (*Fujita et al., 2020*; *Judd et al., 2021*; *Bagnall et al., 2009*). This raises the possibility that the indirect pathway through the deep cerebellar nuclei has the added flexibility of providing both excitation and inhibition.

## Lack of significant PC inputs to LC neurons

We find that PCs do not provide a significant direct input to the LC. This has been controversial because retrograde labeling studies using rabies viruses concluded that PCs constitute one of the largest groups of cells projecting to the LC (*Schwarz et al., 2015*; *Figure 2h and i*), even though PCs did not appear to innervate the core of the LC (*Chen et al., 2023* and *Figures 1 and 2*). We directly assessed PC to LC connection strengths using a cre line that is highly selective for PCs, and we found that in PC/ChR2-YFP mice, tiny optically evoked PC-IPSCs (~50 pA) were observed in just 3% of LC neurons. The average IPSC size for all cells was less than 2 pA. This indicates that within the LC, inhibition originates from brain regions other than PCs (*Breton-Provencher and Sur, 2019*). The cerebellum may still regulate firing in the LC, but it does so via connections through the deep cerebellar nuclei and the LC (*Carlson et al., 2021*; *Cedarbaum and Aghajanian, 1978*; *Aston-Jones et al., 1986*).

Together, these findings indicate that reliance on anatomical tracing experiments alone is insufficient to establish the presence and importance of a synaptic connection. Studies highlighting PC inputs to the LC were conducted using rabies-tracing methods (*Schwarz et al., 2015*; *Sun et al., 2020*; *Breton-Provencher and Sur, 2019*), which have limitations. These viruses have unknown tropism, unclear synapse specificity, and critically depend on the degree to which the 'starter' cell population is restricted to the intended cell type, which can be complicated by inappropriate expression, and by the sensitivity of TVA-based rabies infection (*Beier, 2021*; *Rogers and Beier, 2021*). We were fortunate to have a highly selective cre line to selectively target PCs that allowed us to examine PC synapses in the brainstem while avoiding complications that can occur when using viruses.

## PC targets in the brainstem

The next step in clarifying the PC to brainstem outputs and the behaviors they regulate is to determine the molecular identity of target cells. This is challenging because there are many types of neurons present in the brainstem (*Nardone et al., 2024*; *Langlieb et al., 2023*; *Zhang et al., 2023*; *Kodama et al., 2012*). We previously identified candidate PC targets by injecting AAV1 in the cerebellar cortex to anterogradely label PC targets and then using RNA-seq to determine the molecular identity of labeled cells (*Chen et al., 2023*). An important caveat to this approach is that while it very effectively labels anterograde PC targets, it can also retrogradely label cells that project to the cerebellar cortex. It is therefore crucial to determine if functional PC synapse is present, unless it is known that the region does not project to the cerebellar cortex.

Our primary focus in our previous study (*Chen et al., 2023*) was to identify PC targets in the PB. The fact that the PB does not project to the posterior vermis of the cerebellum (*Huang et al., 2021a*; *Huang et al., 2021b*) and the observation that labeled cells were restricted to PB regions receiving PC inputs suggest that the following cells are PC targets: *Satb2*, *Penk*, and *Tacr1* populations, which have roles in taste (*Fu et al., 2019*; *Jarvie et al., 2021*), thermoregulation (*Norris et al., 2021*), and pain (*Barik et al., 2021*; *Deng et al., 2020*), respectively. We also found that *Calca* neurons (*Huang et al., 2021b*; *Campos et al., 2018*; *Palmiter, 2018*), which are located in a region of the PB devoid of PC synapses, were not labeled by this method. This supports the hypothesis that PCs selectively inhibit specific types of brainstem neurons.

We also labeled neurons in the vestibular nucleus, Barrington's nucleus, mesencephalic trigeminal nucleus, and the LC in that previous study. The lack of PC-evoked inputs to neurons in the mesencephalic trigeminal nucleus and LC shown here established that these populations are not targeted by PCs. It further suggests that these neurons project to the cerebellum and were retrogradely labeled (*Billig et al., 1995*; *Stanley et al., 2023*). For the multiple candidate PC targets in the vestibular nucleus, it is necessary to specifically test for the presence of PC synapses, as was done for floccular-targeting, *Slc6a5*-expressing neurons (*Shin et al., 2011*; *Sekirnjak et al., 2003*). In the PCG and nearby nuclei, the sparsity of PC synapses combined with the large PC synaptic inputs in a small fraction of cells suggests that PCs only inhibit specific cells in these regions. This was also reflected in our previous anatomical experiments (*Chen et al., 2023*), where we identified *Crh*-expressing Barrington's neurons and other glutamatergic/GABAergic neurons in the PCG. These cells are important for micturition (*Hou et al., 2016*) and valence (*Xiao et al., 2023*) respectively, and further indicate a broad range of PC output functions.

# Methods

**Key resources table**

| Reagent type (species) or resource | Designation | Source or reference | Identifiers | Additional information |
|---|---|---|---|---|
| Strain, strain background (C57BL/6) | C57BL/6J | Jackson Laboratory | JAX #000664; RRID:IMSR_JAX:000664 | |
| Strain, strain background (C57BL/6) | B6.Cg-Tg(Pcp2-cre)3555Jdhu/J | Jackson Laboratory | JAX #010536; RRID:IMSR_JAX:010536 | |
| Strain, strain background (C57BL/6) | B6;129S-Gt(ROSA)26Sor<sup>tm34.1(CAG-Syp/tdTomato)Hze</sup>/J | Jackson Laboratory | JAX #012570; RRID:IMSR_JAX:012570; Ai34D | |
| Strain, strain background (C57BL/6) | B6.Cg-Gt(ROSA)26Sor<sup>tm32(CAG-COP4*H134R/EYFP)Hze</sup>/J | Jackson Laboratory | JAX #024109; RRID:IMSR_JAX:024109; Ai32 | |
| Strain, strain background (Swiss Webster) | Tg(Calca-EGFP)FG104 Gsat/Mmucd | MMRRC | 011187-UCD; RRID:MMRRC_011187-UCD | |
| Chemical compound, drug | NBQX disodium salt | Abcam | Ab120046 | |
| Chemical compound, drug | (R)-CPP | Abcam | Ab120159 | |
| Chemical compound, drug | Biocytin | Thermo Fisher | B1592 | |

*Continued on next page*

*Continued*

| Reagent type (species) or resource | Designation | Source or reference | Identifiers | Additional information |
|---|---|---|---|---|
| Antibody | Anti-tyrosine hydroxylase | Sigma-Aldrich | AB152 | Rabbit polyclonal; 1:1000 |
| Antibody | Anti-parvalbumin | Sigma-Aldrich | P3088 | Mouse monoclonal; 1:1000 |
| Antibody | Goat Anti-Rabbit Alexa Fluor 647 secondary | Thermo Fisher | A32733 | Goat anti-rabbit polyclonal; 1:1000 |
| Antibody | Goat Anti-Mouse Alexa Fluor 647 | Thermo Fisher | A21241 | Goat anti-mouse polyclonal; 1:1000 |
| Antibody | Anti-VGAT | Synaptic Systems | 131 004 | Guinea pig polyclonal; 1:500 |
| Antibody | Goat Anti-Guinea Pig Alexa Fluor 647 | Thermo Fisher | A21450 | Goat anti-guinea pig polyclonal; 1:1000 |
| Chemical compound, drug | Streptavidin, Alexa Fluor 594 Conjugate | Thermo Fisher | S11227 | |
| Software, algorithm | Arivis | Zeiss | | https://www.zeiss.com/microscopy/us/products/software/arivis-pro.html |
| Software, algorithm | Igor Pro 8 | WaveMetrics | RRID:SCR_000325 | https://www.wavemetrics.com/ |
| Software, algorithm | SutterPatch | Sutter | | https://www.sutter.com/AMPLIFIERS/SutterPatch.html |
| Software, algorithm | MafPC | Courtesy of MA Xu-Friedman | | https://www.xufriedman.org/mafpc |
| Software, algorithm | MATLAB (R2023b) | MathWorks | RRID:SCR_001622 | https://www.mathworks.com/products/matlab.html |

## Animals

For anatomical studies, B6.Cg-Tg(Pcp2-cre)3555Jdhu/J (JAX #010536 [*Zhang et al., 2004*]) × B6;129S-Gt(ROSA)26Sor$^{tm34.1(CAG-Syp/tdTomato)Hze}$/J (JAX #012570) (PC/synaptophysin-tdTomato, Jackson Laboratory) was used. Tg(Calca-EGFP)FG104Gsat/Mmucd (*Calca$^{GFP}$* mice) were obtained from the Mutant Mouse Resource and Research Center (MMRRC) at University of California at Davis, an NIH-funded strain repository, and was donated to the MMRC by Nathaniel Heintz, Ph.D., The Rockefeller University, GENSAT (RRID:MMRRC_011187-UCD; *Gong et al., 2003*), and generously provided to us by David Ginty. CGRP-GFP mice were crossed with the PC/synaptophysin-tdTomato mice to further support anatomical studies. For physiology experiments, B6.Cg-Tg(Pcp2-cre)3555Jdhu/J (JAX #010536 [*Zhang et al., 2004*]) × B6;129S-Gt(ROSA)26Sor$^{tm32(CAG-COP4*H134R/EYFP)Hze}$/J (JAX #024109; *Madisen et al., 2012*) (PC/ChR2-YFP, Jackson Laboratory) mice were used. All animal procedures were carried out in accordance with the NIH and the Institutional Animal Care and Use Committee (IACUC) guidelines and protocols approved by the Harvard Medical Area Standing Committee on Animals (protocol #IS00000124) or Pennsylvania State University College of Medicine IACUC (protocol #PROTO202202365). Mice from either sex from Jackson Laboratory were used for all experiments.

## Anatomy

In order to identify inhibitory synapses PCs made in the brainstem, PC/synaptophysin-tdTomato mice were crossed with *Calca$^{GFP}$* mice. Thus, synaptophysin-tdTomato was expressed selectively in all PCs, and GFP is expressed in the lateral parabrachial nuclei, the LC, the hypoglossal nuclei, the vestibular nerve, the spinal tract of the trigeminal nerve, and the solitary tracts. Mice were anesthetized at P56 and perfused with PBS and 4% PFA. After removal, brains were post-fixed for 1 day in 4% PFA, and sliced coronally and sagittally in 50 µm sections. Slices were stained with a vGAT antibody

(Synaptic Systems, #131 004, 1:500), and visualized using an Alexa Fluor 647 secondary (Thermo Fisher, #A-21450, 1:1000). Every third section of the dorsal brainstem was imaged on a confocal microscope (Leica Stellaris 5). Acquisitions were made with an XY resolution of 361.1 nm for an area of about 10 mm² total per section, and 300 nm z steps for 4 µm stacks. Stitched images were imported into ZEISS arivis Pro (arivis).

To identify PC synapses, we analyzed for synaptophysin-tdTomato fluorescence colabeled with vGAT fluorescence (*Chen et al., 2023*; *Guo et al., 2021*). Using a background correction feature in arivis, a background representation was generated with a 50 µm diameter Gaussian filter and subtracted from the vGAT fluorescence. vGAT was then isolated by a watershed algorithm provided with a 1 µm bouton diameter and threshold intensity. Thresholds were adjusted for each individual image based on three 50-pixel squares that had various densities of vGAT expression. Optimal threshold values were approximately one standard deviation above median. The following analysis was conducted in MATLAB: segmented vGAT boutons were analyzed for tdTomato-synaptophysin colocalization by finding a regional maximum in the tdTomato fluorescence (maximum center-to-center distance is 0.7 µm). We plotted the synaptophysin intensity values within each voxel. This histogram showed a bimodal distribution corresponding to noise and signal, and the signal threshold corresponded to the minimum between the two peaks.

We identified brainstem nuclei by aligning the Allen CCF. We used several features to register confocal images of the brainstem: (1) the edges of the tissue and position of the ventricles, (2) the fiber tracts and features notable from autofluorescence and bright-field imaging, and (3) GFP labeling present in numerous brainstem regions in *Calca^GFP* mice (including VIIn, sptv, LC, XII, and their axons). GFP labeling in the LC and VIIn was particularly useful in anterior sections of *Calca^GFP* mice.

Puncta within each brainstem region were quantified using this aligned atlas. We matched our samples to the indicated bregma levels (±100 µm) in *Figure 2*. To determine the variability in our alignment, we sampled 50–70 points 350 µm apart across the atlas image before and after aligning to each confocal image. The displacement distances of these reference points were calculated for each image and binned based on the confocal image location relative to bregma. Our images were displaced from the Allen CCF by an average of 107 µm. The displacement distances were 90±25 µm (standard deviation, n=6 hemispheres, bregma –5.45 mm), 126±26 µm (n=10, bregma –5.65 mm), 102±35 µm (n=8, bregma –6.05 mm), 103±51 µm (n=8, bregma –6.35 mm), 108±43 µm (n=8, bregma –6.65 mm), and 96±31 µm (n=8, bregma –6.95 mm).

It is important to note that the accuracy of our mapping is dependent on the accuracy of the reference maps that we register to. The PCG is one region in particular that varies substantially between reference atlases (*Wang et al., 2020*; *Paxinos and Franklin, 2019*; *Kronman et al., 2023*). We labeled the regions in/around the dorsomedial pons as the PCG, based off of the borders noted in the Allen CCF, but other atlases constrain the borders of the PCG, leaving parts of the dorsomedial pons effectively unlabeled. The precise location of many other regions within the dorsomedial pons also differs dramatically. Precise identification of these areas (i.e. Barrington's nucleus) will require a molecularly targeted approach (in situ hybridization, etc.).

## Electrophysiology

Mice of both sexes ranging from p24 to p50 (n=30) were anesthetized with ketamine/xylazine and transcardially perfused with warm choline ACSF solution (34°C) containing in mM: 110 choline Cl, 2.5 KCl, 1.25 NaH₂PO₄, 25 NaHCO₃, 25 glucose, 0.5 CaCl₂, 7 MgCl₂, 3.1 Na pyruvate, 11.6 Na ascorbate, 0.002 (R,S)-CPP, 0.005 NBQX, oxygenated with 95% O₂/5% CO₂. Coronal slices were made (200 µm) using a Leica 1200S or a Campden Instrument 7000smz-2 vibratome in warm choline ACSF (34°C). Slices were then transferred to a holding chamber with warm ACSF (34°C) containing in mM: 127 NaCl, 2.5 KCl, 1.25 NaH₂PO₄, 25 NaHCO₃, 25 glucose, 1.5 CaCl₂, 1 MgCl₂ and were recovered at 34°C for 30 min before being moved to room temperature until recordings began.

Voltage-clamp recordings were made across the brainstem. Borosilicate glass electrodes (2–4 MΩ) were filled with a high chloride internal containing in mM: 35 CsF, 100 CsCl, 10 EGTA, 10 HEPES, 4 QX-314, pH 7.3 with CsOH. This internal solution was used to increase sensitivity to inhibitory inputs and improve clamp. The osmolarity of internal solution was adjusted to 290–300 mOsm. Whole-cell capacitance and series resistance was left uncompensated for all experiments. Neurons were held at –70 mV. All experiments were performed at room temperature in the presence of 5 µM NBQX to block

AMPARs and 2.5 µM (R,S)-CPP to block NMDARs, with a flow rate of 2 ml/min. ChR2 expressing PC axons were stimulated with 473 nm light from a light-emitting diode (Thorlabs or Sutter) in the entire field of view (1 ms, ~80 mW/mm$^2$) for at least 10 trials. Data were collected using Igor (WaveMetrics) running mafPC (courtesy of MA Xu-Friedman) or SutterPatch (Sutter). Images with a ×4 objective were used to determine the positions of recorded cells. Slices were fixed and stained with DAPI to confirm the anterior-posterior position of the coronal slice, and matched to the closest indicated bregma level indicated in *Figure 3*.

In experiments targeting the LC or mesencephalic trigeminal nucleus, biocytin (B1592, Thermo Fisher) was included in the intracellular solution. Cells were then stained with an anti-tyrosine hydroxylase antibody to identify the LC (1:1000, AB152, Sigma-Aldrich) or anti-parvalbumin antibody to identify the mesencephalic trigeminal (1:1000 P3088, Sigma-Aldrich), followed by an Alexa Fluor 647 secondary antibody (1:1000 A32733 or A21241, Thermo Fisher, respectively) and Streptavidin, Alexa Fluor 594 Conjugate (1:1000, S11227, Thermo Fisher). Sections were subsequently imaged using a confocal microscope (Leica Stellaris 5) to determine the identity of the recorded cell.

## Acknowledgements

This work was supported by grants from the National Institutes of Health: R01NS032405 and R35NS097284 to WGR, R00NS110978 to CHC, and NINDS P30 Core Center (NS072030) to the Neurobiology Imaging Center at Harvard Medical School. Shuting Wu was supported by the Lefler Center at Harvard Medical School.

## Additional information

### Funding

| Funder | Grant reference number | Author |
|---|---|---|
| National Institute of Mental Health | R01NS032405 | Wade G Regehr |
| National Institute of Neurological Disorders and Stroke | R35NS097284 | Wade G Regehr |
| National Institute of Neurological Disorders and Stroke | R00NS110978 | Christopher H Chen |
| Lefler Center | | Shuting Wu |

The funders had no role in study design, data collection and interpretation, or the decision to submit the work for publication.

### Author contributions

Christopher H Chen, Conceptualization, Data curation, Supervision, Funding acquisition, Writing – original draft, Writing – review and editing; Zhiyi Yao, Data curation, Investigation, Visualization, Writing – review and editing; Shuting Wu, Investigation, Writing – review and editing; Wade G Regehr, Conceptualization, Funding acquisition, Writing – original draft, Project administration, Writing – review and editing

### Author ORCIDs

Christopher H Chen https://orcid.org/0000-0002-4611-8667
Shuting Wu https://orcid.org/0000-0003-2264-7393
Wade G Regehr https://orcid.org/0000-0002-3485-8094

### Ethics

All animal procedures were carried out in accordance with the NIH and the Institutional Animal Care and Use Committee (IACUC) guidelines and protocols approved by the Harvard Medical Area Standing Committee on Animals (protocol #IS00000124) or Pennsylvania State University College of

Medicine IACUC (protocol #PROTO202202365). Mice from either sex from Jackson Labs were used for all experiments.

Reviewer #1 (Public review): https://doi.org/10.7554/eLife.101825.3.sa1
Reviewer #2 (Public review): https://doi.org/10.7554/eLife.101825.3.sa2
Reviewer #3 (Public review): https://doi.org/10.7554/eLife.101825.3.sa3
Author response https://doi.org/10.7554/eLife.101825.3.sa4

---

## Additional files

### Supplementary files
MDAR checklist

### Data availability
All data that supports the findings in this study have been deposited in Dryad.

The following dataset was generated:

| Author(s) | Year | Dataset title | Dataset URL | Database and Identifier |
|---|---|---|---|---|
| Yao Z, Chen C, Wu S, Regehr W | 2025 | Data from: Characterization of direct Purkinje cell outputs to the brainstem | https://doi.org/10.5061/dryad.s7h44j1hw | Dryad Digital Repository, 10.5061/dryad.s7h44j1hw |

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
