## [Editor Report · eLife Assessment]

This **important** paper provides an unbiased landscape for the cerebellar cortical outputs to the brainstem nuclei. By conducting anatomical and physiological analyses of the axonal terminals of Purkinje cells, the data provides **convincing** evidence that Purkinje cells innervate brainstem nuclei directly. The results show that in addition to previously known inputs to vestibular and parabrachial nuclei, Purkinje cells synapse onto the pontine central grey nucleus but have little effect on the locus coeruleus and mesencephalic trigeminal neurons.

---

## [Referee Report · Reviewer #1 (Public review)]

Summary:

This paper is an incremental follow-up to the authors' recent paper which showed that Purkinje cells make inhibitory synapses onto brainstem neurons in the parabrachial nucleus which project directly to the forebrain. In that precedent paper, the authors used a mouse line which expresses the presynaptic marker synaptophysin in Purkinje cells to identify Purkinje cell terminals in the brainstem and they observed labeled puncta not only in the vestibular and parabrachial nuclei, as expected, but also in neighboring dorsal brainstem nuclei, prominently the central pontine grey. The present study, motivated by the lack of thorough characterization of PC projections to brainstem, uses the same mouse line to anatomically map the density and a PC-specific channelrhodopsin mouse line to electrophysiologically assess the strength of Purkinje cell synapses in dorsal brainstem nuclei. The main findings are (1) the density of Purkinje cell synapses is highest in vestibular and parabrachial nuclei and correlates with the magnitude of evoked inhibitory synaptic currents, and (2) Purkinje cells also synapse in the central pontine grey nucleus but not in the locus coeruleus or mesencephalic nucleus.

Strengths:

The complementary use of anatomical and electrophysiological methods to survey the distribution and efficacy of Purkinje cell synapses on brainstem neurons in mouse lines that express markers and light-sensitive opsins specifically in Purkinje cells is the major strength of this study. By systematically mapping presynaptic terminals and light-evoked inhibitory postsynaptic currents in dorsal brainstem, the authors provide convincing evidence that Purkinje cells do synapse directly onto pontine central grey and nearby neurons but do not synapse onto trigeminal motor or locus coeruleus neurons. Their results also confirm previously documented heterogeneity of Purkinje cell inputs to vestibular nucleus and parabrachial neurons.

Weaknesses:

Although the study provides strong evidence that Purkinje cells do not make extensive synapses onto LC neurons, which is a helpful caveat given previous reports to the contrary, it falls short of providing the comprehensive characterization of Purkinje cell brainstem synapses which seemed to be the primary motivation of the study. The main information provided is a regional assessment of PC density and efficacy, which seems of limited utility given that we are not informed about the different sources of PC inputs, variations in the sizes of PC terminals, the subcellular location of synaptic terminals, or the anatomical and physiological heterogeneity of postsynaptic cell types. The title of this paper would be more accurate if "characterization" were replaced by "survey".

Several of the study's conclusions are quite general and have already been made for vestibular nuclei, including the suggestions in Abstract, Results, and Discussion that PCs selectively influence brainstem subregions and that PCs target cell types with specific behavioral roles.

---

## [Referee Report · Reviewer #2 (Public review)]

Summary:

While it is often assumed that the cerebellar cortex connects, via its sole output neuron, Purkinje cell, exclusively to the cerebellar nuclei, axonal projections of the Purkinje cells to dorsal brainstem regions have been well documented. This paper provides comprehensive mapping and quantification of such extracerebellar projections of the Purkinje cells, most of which are confirmed with electrophysiology in slice preparation. A notable methodological strength of this work is the use of highly Purkinje cell-specific transgenic strategies, enabling selective and unbiased visualization of Purkinje terminals in the brainstem. By utilizing these selective mouse lines, the study offers compelling evidence challenging the general assumption that Purkinje cell targets are limited to the cerebellar nuclei. While the individual connections presented are not entirely novel, this paper provides a thorough and unambiguous demonstration of their collective significance. Regarding another major claim of this paper, "characterization of direct Purkinje cell outputs (Title)", however, the depth of electrophysiological analysis is limited to presence/absence of physiological Purkinje input to postsynaptic brainstem neurons whose known cell types are mostly blinded. Overall, conceptual advance is largely limited to confirmatory or incremental, although it would be useful for the field to have the comprehensive landscape presented.

Strengths:

Unsupervised comprehensive mapping and quantification of the Purkinje terminals in the dorsal brainstem are enabled, for the first time, by using the current state-of-the-art mouse lines, BAC-Pcp2-Cre and synaptophysin-tdTomato reporter (Ai34).

Combinatorial quantification with vGAT puncta and synaptophysin-tdTomato labeled Purkinje terminals clarifies the anatomical significance of the Purkinje terminals as an inhibitory source in each dorsal brainstem region.

Electrophysiological confirmation of the presence of physiological Purkinje synaptic input to 7 out of 9 dorsal brainstem regions identified.

Pan-Purkinje ChR2 reporter provides solid electrophysiological evidence to help understand the possible influence of the Purkinje cells onto LC.

Weaknesses:

The present paper is largely confirmatory to what is presented in a previous paper published by the author's group (Chen et al., 2023, Nat Neurosci). In this preceding paper, the author's group used AAV1-mediated anterograde transsynaptic strategy to identify postsynaptic neurons of the Purkinje cells. The experiments performed in the present paper is, by nature, complementary to the AAV1 tracing which can also infect retrogradely and thus is not able to demonstrate the direction of synaptic connections between reciprocally connected regions. Anatomical findings are all consistent with the preceding paper.

While the authors appear to assume uniform cell type and postsynaptic response in each of the dorsal brainstem nuclei (as noted in the Discussion, "PCs likely function similarly to their inputs to the cerebellar nuclei, where a very brief pause in firing can lead to large and rapid elevations in target cell firing"), we know that the responses to the Purkinje cell input are cell type dependent, which vary in neurotransmitter, output targets, somata size, and distribution, in the cerebellar and vestibular nuclei (Shin et al., 2011, J Neurosci; Najac and Raman, 2015, J Neurosci; Özcan et al., 2020, J Neurosci). Also, whether 23 % (for PCG), for example, is "a small fraction" would be subjective: it might represent a numerically small but functionally important cell type population. From a functional perspective, the cell type-blind physiological characterization in this manuscript remains superficial compared to existing cell type-specific analyses, although the authors commented on these issues in the manuscript.

---

## [Referee Report · Reviewer #3 (Public review)]

Summary:

The manuscript by Chen and colleagues explores the connections from cerebellar purkinje cells to various brainstem nuclei. They combine two methods - presynaptic puncta labeling as putative presynaptic markers, and optogenetics, to test the anatomical projections and functional connectivity from purkinje cells onto a variety of brainstem nuclei. Overall, their study provides an atlas of sorts of purkinje cell connectivity to the brainstem, which includes a critical analysis of some of their own data from another publication. Overall, the value of this work is to both provide neural substrates by which purkinje cells may influence the brainstem and subsequent brain regions independent of the deep cerebellar nuclei, and also, to provide a critical analysis of viral-based methods to explore neuronal connectivity.

Strengths:

The strengths lie in the simplicity of the study, the number of cells patched, and the relationship between the presence of putative presynaptic puncta and electrophysiological results. This type of study is important and should provide a foundation for future work exploring cerebellar inputs and outputs. Overall, I think that the critique of viral-based methods to define connectivity, and a more holistic assessment of what connectivity is and how it should be defined is timely and warranted, as I think this is under-appreciated by many groups and overall, there is a good deal of research being published that do not properly consider the issues that this manuscript raises about what viral-based connectivity maps do and do not tell us.

Weaknesses:

While I overall liked the manuscript, I do have a few concerns which relate to interpretation of results, and discussion of technological limitations. The main concerns I have relate to the techniques that the authors use, and an insufficient discussion of their limitations. The authors use a Cre-dependent mouse line that expresses a synaptophysin-tdtomato marker, which the authors confidently state is a marker of synapses. This is misleading. Synaptophysin is a vesicle marker, and as such, labels axons, where vesicles are present in transit, and likely cell bodies where the protein is being produced. As such, the presence of tdtomato should not be interpreted definitively as the presence of a synapse. The use of vGAT as a marker, while this helps to constrain the selection of putative pre-synaptic sites, is also a vesicle marker and will likely suffer the same limitations (though in this case the expression is endogenous and not driven by the ROSA locus). A more conservative interpretation of the data would be that the authors are assessing putative pre-synaptic sites with their analysis. This interpretation is wholly consistent with their findings showing the presence of tdtomato in some regions but only sparse connectivity - this would be expected in the event that axons are passing through. If the authors wish to strongly assert that they are specifically assessing synapses, a marker better restricted to synapses and not vesicles may be more appropriate.

Similarly, while optogenetics/slice electrophysiology remains the state of the art for assessing connectivity between cell populations, it is not without limitations. For example, connections that are not contained within the thickness of the slice (here, 200 um, which is not particularly thick for slice ephys preps) will not be detected. As such, the absence of connections are harder to interpret than the presence of connections. Slices were only made in the coronal plane, which means if that if there is a particular topology to certain connections that is orthogonal to that plane, those connections may be under-represented. As such, all connectivity analyses likely are under-representations of the actual connectivity that exists in the intact brain. Therefore, perhaps the authors should consider revising their assessments of connections, or lack thereof, of purkinje cells to e.g., LC cells. While their data do make a compelling case that the connections between purkinje cells and LC cells are not particularly strong or numerous, especially compared to other nearby brainstem nuclei, their analyses do indicate that at least some such connections do exist. Thus, rather than saying that the viral methods such as rabies virus are not accurate reflections of connectivity - perhaps a more circumspect argument would be that the quantitative connectivity maps reported by other groups using rabies virus do not always reflect connectivity defined by other means e.g., functional connections with optogenetics. In some cases the authors do suggest this (e.g., "Together, these findings indicate that reliance on anatomical tracing experiments alone is insufficient to establish the presence and important of a synaptic connection"), but in other cases they are more dismissive of viral tracing results (e.g., "it further suggests that these neurons project to the cerebellum and were not retrogradely labeled"). Furthermore, some statements are a bit misleading e.g., mentioning that rabies methods are critically dependent on starter cell identity immediately following the citation of studies mapping inputs onto LC cells. While in general this claim has merit, the studies cited (19-21) use Dbh-Cre to define LC-NE cells which does have good fidelity to the cells of interest in the LC. Therefore, rewording this section in order to raise these issues generally without proximity to the citations in the previous sentence may maintain the authors' intention without suggesting that perhaps the rabies studies from LC-NE cells that identified inputs from purkinje cells were inaccurate due to poor fidelity of the Cre line. Overall, this manuscript would certainly not be the first report indicating that rabies virus does not provide a quantitative map of input connections. In my opinion this is still under-appreciated by the broad community and should be explicitly discussed. Thus, an acknowledgement of previous literature on this topic and how their work contributes to that argument is warranted.

Comments on revisions:

The responses the authors offer in theory are good, but they still use terms such as synapses and putative presynaptic boutons relatively interchangeably - if the authors make the correction to the more conservative terminology, which I think better reflects the data, this should be more consistent throughout the manuscript.

---

## [Author Response]

The following is the authors’ response to the original reviews.

**Public Reviews:**

**Reviewer #1 (Public review):**
Summary:This paper is an incremental follow-up to the authors' recent paper which showed that Purkinje cells make inhibitory synapses onto brainstem neurons in the parabrachial nucleus which project directly to the forebrain. In that precedent paper, the authors used a mouse line that expresses the presynaptic marker synaptophysin in Purkinje cells to identify Purkinje cell terminals in the brainstem and they observed labeled puncta not only in the vestibular and parabrachial nuclei, as expected, but also in neighboring dorsal brainstem nuclei, prominently the central pontine grey. The present study, motivated by the lack of thorough characterization of PC projections to the brainstem, uses the same mouse line to anatomically map the density and a PC-specific channelrhodopsin mouse line to electrophysiologically assess the strength of Purkinje cell synapses in dorsal brainstem nuclei. The main findings are (1) the density of Purkinje cell synapses is highest in vestibular and parabrachial nuclei and correlates with the magnitude of evoked inhibitory synaptic currents, and (2) Purkinje cells also synapse in the central pontine grey nucleus but not in the locus coeruleus or mesencephalic nucleus.Strengths:The complementary use of anatomical and electrophysiological methods to survey the distribution and efficacy of Purkinje cell synapses on brainstem neurons in mouse lines that express markers and light-sensitive opsins specifically in Purkinje cells is the major strength of this study. By systematically mapping presynaptic terminals and light-evoked inhibitory postsynaptic currents in the dorsal brainstem, the authors provide convincing evidence that Purkinje cells do synapse directly onto pontine central grey and nearby neurons but do not synapse onto trigeminal motor or locus coeruleus neurons. Their results also confirm previously documented heterogeneity of Purkinje cell inputs to the vestibular nucleus and parabrachial neurons.Weaknesses:Although the study provides strong evidence that Purkinje cells do not make extensive synapses onto LC neurons, which is a helpful caveat given previous reports to the contrary, it falls short of providing the comprehensive characterization of Purkinje cell brainstem synapses which seemed to be the primary motivation of the study. The main information provided is a regional assessment of PC density and efficacy, which seems of limited utility given that we are not informed about the different sources of PC inputs, variations in the sizes of PC terminals, the subcellular location of synaptic terminals, or the anatomical and physiological heterogeneity of postsynaptic cell types. The title of this paper would be more accurate if "characterization" were replaced by "survey".Several of the study's conclusions are quite general and have already been made for vestibular nuclei, including the suggestions in the Abstract, Results, and Discussion that PCs selectively influence brainstem subregions and that PCs target cell types with specific behavioral roles.

We agree that we did not provide an in-depth characterization of PC synapses onto all identified types of brainstem neurons. With so many types of neurons in the brainstem, this would be a monumental task. Despite this limitation we prefer to keep our original title, since our study makes the following advances:

• We provide a comprehensive map of all PC synaptic boutons across the brainstem, and corresponding maps of PC synaptic input sizes. The input sizes vary widely, but are often multiple nanoamps, indicating that the cerebellum is an important regulator of activity in these regions. These maps will be indispensable for future investigations of cerebellar outputs.

• We find that PC projections and the synapses they make are spatially restricted within most target nuclei such as the vestibular and parabrachial nuclei. This suggests that the influence of the cerebellum is spatially segregated within these nuclei, and likely allows the cerebellum to regulate specific behaviors. While some aspects of these gradients have been described previously, our study is comprehensive, and has a higher degree of specificity than can be achieved with immunohistochemistry.

• We discover that PCs form functional synapses in the pontine central grey and nearby nuclei. Much of this region’s function is unknown, but certain subregions are important for micturition and valence. PCs make large synapses onto a small fraction of cells in this region, which suggests that PCs may target specific cell types to control novel nonmotor behaviors.

• We provide clarification regarding PC projections to the locus coeruleus. Multiple high-profile, highly influential studies using rabies tracing (Schwarz et al., *Nature* 2015; Breton-Provencher and Sur, *Nature Neuroscience* 2019; and others) described a prominent PC input to the locus coeruleus. We showed that this projection is essentially nonexistent, both anatomically and functionally. We previously addressed this issue, but the PC-specific optogenetic approach we used here provides the most compelling evidence against a prominent PC-LC connection. This is an important finding for the cerebellum and a cautionary tale for conclusions based solely on viral tracing methods. We will expand on this issue in response to the comments of reviewer #3.

**Reviewer #2 (Public review):**
Summary:While it is often assumed that the cerebellar cortex connects, via its sole output neuron, the Purkinje cell, exclusively to the cerebellar nuclei, axonal projections of the Purkinje cells to dorsal brainstem regions have been well documented. This paper provides comprehensive mapping and quantification of such extracerebellar projections of the Purkinje cells, most of which are confirmed with electrophysiology in slice preparation. A notable methodological strength of this work is the use of highly Purkinje cell-specific transgenic strategies, enabling selective and unbiased visualization of Purkinje terminals in the brainstem. By utilizing these selective mouse lines, the study offers compelling evidence challenging the general assumption that Purkinje cell targets are limited to the cerebellar nuclei. While the individual connections presented are not entirely novel, this paper provides a thorough and unambiguous demonstration of their collective significance. Regarding another major claim of this paper, "characterization of direct Purkinje cell outputs (Title)", however, the depth of electrophysiological analysis is limited to the presence/absence of physiological Purkinje input to postsynaptic brainstem neurons whose known cell types are mostly blinded. Overall, conceptual advance is largely limited to confirmatory or incremental, although it would be useful for the field to have the comprehensive landscape presented.Strengths:(1) Unsupervised comprehensive mapping and quantification of the Purkinje terminals in the dorsal brainstem are enabled, for the first time, by using the current state-of-the-art mouse lines, BAC-Pcp2-Cre and synaptophysin-tdTomato reporter (Ai34).(2) Combinatorial quantification with vGAT puncta and synaptophysin-tdTomato labeled Purkinje terminals clarifies the anatomical significance of the Purkinje terminals as an inhibitory source in each dorsal brainstem region.(3) Electrophysiological confirmation of the presence of physiological Purkinje synaptic input to 7 out of 9 dorsal brainstem regions identified.(4) Pan-Purkinje ChR2 reporter provides solid electrophysiological evidence to help understand the possible influence of the Purkinje cells onto LC.Weaknesses:(1) The present paper is largely confirmatory of what is presented in a previous paper published by the author's group (Chen et al., 2023, Nat Neurosci). In this preceding paper, the author's group used AAV1-mediated anterograde transsynaptic strategy to identify postsynaptic neurons of the Purkinje cells. The experiments performed in the present paper are, by nature, complementary to the AAV1 tracing which can also infect retrogradely and thus is not able to demonstrate the direction of synaptic connections between reciprocally connected regions. Anatomical findings are all consistent with the preceding paper. The likely absence of robust physiological connections from the Purkinje to LC has also been evidenced in the preceding paper by examining c-Fos response to Purkinje terminal photoinhibition at the PBN/LC region.

We agree that we previously dealt with the issue of PC-LC synapses (Chen et al., 2023, Nat Neurosci), but our conclusions differed from several high-profile publications (Schwarz et al., *Nature* 2015; Breton-Provencher and Sur, *Nature Neuroscience* 2019), and still met considerable resistance. We felt that the optogenetic approach provided the most definitive means of evaluating the presence and strength of PC-LC synapse that will hopefully settle this issue. These experiments also set a standard for future studies assessing the presence of PC synapses onto other target neurons in the brainstem.

(2) Although the authors appear to assume uniform cell type and postsynaptic response in each of the dorsal brainstem nuclei (as noted in the Discussion, "PCs likely function similarly to their inputs to the cerebellar nuclei, where a very brief pause in firing can lead to large and rapid elevations in target cell firing"), we know that the responses to the Purkinje cell input are cell type dependent, which vary in neurotransmitter, output targets, somata size, and distribution, in the cerebellar and vestibular nuclei (Shin et al., 2011, J Neurosci; Najac and Raman, 2015, J Neurosci; Özcan et al., 2020, J Neurosci). This consideration impacts the interpretation of two key findings: (a) "Large ... PC-IPSCs are preferentially observed in subregions with the highest densities of PC synapses (Abstract)". For example, we know that the terminal sparse regions reported in the present paper do contain Floccular Targeted Neurons that are sparse yet have dense somatic terminals with profound postinhibitory rebound (Shin et al.). Despite their sparsity, these postsynaptic neurons play a distinct and critical role in proper vestibuloocular reflex. Therefore, associating broad synaptic density with "PC preferential" targets, as written in the Abstract, may not fully capture the behavioral significance of Purkinje extracerebellar projections. (b) "We conclude ... only a small fraction of cell. This suggests that PCs target cell types with specific behavioral roles (Abstract, the last sentence)". Prior research has already established that "PCs target cell types with specific behavioral roles in brainstem regions". Also, whether 23 % (for PCG), for example, is "a small fraction" would be subjective: it might represent a numerically small but functionally important cell type population. The physiological characterization provided in the present cell type-blind analysis could, from a functional perspective, even be decremental when compared to existing cell typespecific analyses of the Purkinje cell inputs in the literature.

We now cite the papers suggested by the reviewer (Shin et al., 2011, J Neurosci; Najac and Raman, 2015, J Neurosci; Özcan et al., 2020, J Neurosci) and add to the discussion.

(3) The quantification analyses used to draw conclusions about(a) the significance of PC terminals among all GABAergic terminals and the fractions of electrophysiologically responsive postsynaptic brainstem neurons may have potential sampling considerations:.(a.i) this study appears to have selected subregions from each brainstem nucleus for quantification (Figure 2). However, the criteria for selecting these subregions are not explicitly detailed, which could affect the interpretation of the results.

Additional explanation has been added to results in the section, “Quantification of PC synapses in the brainstem.”

(a.ii) the mapping of recorded cells (Figure 3) seems to show a higher concentration in terminal-rich regions of the vestibular nuclei.

In Figure 3, we strived to record in an unbiased manner. However, there may have been a slight bias to recordings in areas of lower myelination where patching is easier. We now clarify this issue in the text.

**Reviewer #3 (Public review):**
Summary:The manuscript by Chen and colleagues explores the connections from cerebellar Purkinje cells to various brainstem nuclei. They combine two methods - presynaptic puncta labeling as putative presynaptic markers, and optogenetics, to test the anatomical projections and functional connectivity from Purkinje cells onto a variety of brainstem nuclei. Overall, their study provides an atlas of sorts of Purkinje cell connectivity to the brainstem, which includes a critical analysis of some of their own data from another publication. Overall, the value of this work is to both provide neural substrates by which Purkinje cells may influence the brainstem and subsequent brain regions independent of the deep cerebellar nuclei and also, to provide a critical analysis of viral-based methods to explore neuronal connectivity.Strengths:The strengths lie in the simplicity of the study, the number of cells patched, and the relationship between the presence of putative presynaptic puncta and electrophysiological results. This type of study is important and should provide a foundation for future work exploring cerebellar inputs and outputs. Overall, I think that the critique of viral-based methods to define connectivity, and a more holistic assessment of what connectivity is and how it should be defined is timely and warranted, as I think this is under-appreciated by many groups and overall, there is a good deal of research being published that do not properly consider the issues that this manuscript raises about what viral-based connectivity maps do and do not tell us.

We thank the reviewer for highlighting this important aspect of this work, and for agreeing with our thesis concerning viral-based connectivity maps.

Weaknesses:While I overall liked the manuscript, I do have a few concerns that relate to interpretation of results, and discussion of technological limitations. The main concerns I have relate to the techniques that the authors use, and an insufficient discussion of their limitations. The authors use a Cre-dependent mouse line that expresses a synaptophysin-tomato marker, which the authors confidently state is a marker of synapses. This is misleading. Synaptophysin is a vesicle marker, and as such, labels axons, where vesicles are present in transit, and likely cell bodies where the protein is being produced. As such, the presence of tdtomato should not be interpreted definitively as the presence of a synapse. The use of vGAT as a marker, while this helps to constrain the selection of putative pre-synaptic sites, is also a vesicle marker and will likely suffer the same limitations (though in this case, the expression is endogenous and not driven by the ROSA locus). A more conservative interpretation of the data would be that the authors are assessing putative pre-synaptic sites with their analysis. This interpretation is wholly consistent with their findings showing the presence of tdtomato in some regions but only sparse connectivity - this would be expected in the event that axons are passing through. If the authors wish to strongly assert that they are specifically assessing synapses, a marker better restricted to synapses and not vesicles may be more appropriate.

We agree that synaptophysin-tdTomato is an imperfect marker, although it is vastly superior to cytosolic tdTomato. We found that viral expression of synaptophysin-GFP gives much more punctate labelling, but an appropriate synaptophysin-GFP line is not available. We carefully point out this issue, and threshold the images to avoid faint labeling associated with fibers of passage. The intersection of VGAT labelling and of the synaptophysin-tdTomato labelling provides us with superior identification of PC boutons. We will add additional clarification to point out that these are putative presynaptic boutons, but that alone this does not establish the existence or the strength of functional synapses.

Similarly, while optogenetics/slice electrophysiology remains the state of the art for assessing connectivity between cell populations, it is not without limitations. For example, connections that are not contained within the thickness of the slice (here, 200 um, which is not particularly thick for slice ephys preps) will not be detected. As such, the absence of connections is harder to interpret than the presence of connections. Slices were only made in the coronal plane, which means that if there is a particular topology to certain connections that is orthogonal to that plane, those connections may be under-represented. As such, all connectivity analyses likely are under-representations of the actual connectivity that exists in the intact brain. Therefore, perhaps the authors should consider revising their assessments of connections, or lack thereof, of Purkinje cells to e.g., LC cells. While their data do make a compelling case that the connections between Purkinje cells and LC cells are not particularly strong or numerous, especially compared to other nearby brainstem nuclei, their analyses do indicate that at least some such connections do exist. Thus, rather than saying that the viral methods such as rabies virus are not accurate reflections of connectivity - perhaps a more circumspect argument would be that the quantitative connectivity maps reported by other groups using rabies virus do not always reflect connectivity defined by other means e.g., functional connections with optogenetics. In some cases, the authors do suggest this (e.g."Together, these findings indicate that reliance on anatomical tracing experiments alone is insufficient to establish the presence and importance of a synaptic connection"), but in other cases, they are more dismissive of viral tracing results (e.g. "it further suggests that these neurons project to the cerebellum and were not retrogradely labeled"). Furthermore, some statements are a bit misleading e.g., mentioning that rabies methods are critically dependent on starter cell identity immediately following the citation of studies mapping inputs onto LC cells. While in general, this claim has merit, the studies cited (19-21) use Dbh-Cre to define LC-NE cells which does have good fidelity to the cells of interest in the LC. Therefore, rewording this section in order to raise these issues generally without proximity to the citations in the previous sentence may maintain the authors' intention without suggesting that perhaps the rabies studies from LC-NE cells that identified inputs from Purkinje cells were inaccurate due to poor fidelity of the Cre line. Overall, this manuscript would certainly not be the first report indicating that the rabies virus does not provide a quantitative map of input connections. In my opinion, this is still under-appreciated by the broad community and should be explicitly discussed. Thus, an acknowledgment of previous literature on this topic and how their work contributes to that argument is warranted.

We have a different take on connectivity and the use of optogenetics. Based on our years of experience studying synapses in brain slice, axons survive very well even when they are cut. It is not necessary to preserve intact axons that extend for long distances. It is also true that activation of these axons, with either extracellular electrical stimulation or with optogenetics, is sufficient to evoke synaptic inputs. Robust synaptic responses are evoked with optogenetic activation regardless of the slice orientation. We thank the reviewer for raising this issue, and we have added a couple of sentences to clarify this point under the section “Characterization of functional properties of PC synapses in the brainstem.”

The discussion on starter cell specificity was not referring to the specificity of cre in transgenic animals, but the TVA/G helper proteins that are introduced by AAV and used in conjunction with the rabies virus. The issues related to this have recently been discussed in Elife (Beier, 2022) in addition to citations 58 and 59 in the manuscript. We have more explicitly highlighted this issue in the revised manuscript in the section “Lack of significant PC inputs to LC neurons.”

**Recommendations for the authors:**

**Reviewer #1 (Recommendations for the authors):**
(1) Methods need detail to be replicable, particularly in how PC synapses were identified and automatically counted. It is not clear what was the variation within subregions across mice. How were neurons selected or rejected for recordings and analyses? Was each subregion sampled at equal spacing? Methods for anatomy should mention sagittal sections.

Wording in Methods section, “Anatomy” was changed to better reflect how PC synapses were identified as colabeled segments of vGAT and tdTomato labeling.

Each datapoint in Figure 2D-F was quantification of a region for each section and each mouse. The color of the data point indicates the anterior posterior location of the section. The violin plot quantifies the median and quartile value for all points across sections and mice. The variability captured by the violin point reflects variability across the anterior-posterior axis.

Neurons were mostly randomly selected in each slice, and rejected based on unstable holding current or access resistance. Cell locations were recorded and updated with each experiment so that we minimized oversampling easier to patch regions.

Sagittal sections were added in methods.

(2) Figure 2D-F what is the black line and grey region?

Additional text was added in the caption for Figure 2D-F

(3) MEV is confusing given LAV stands for lateral vestibular - perhaps call it ME5?

We will remain consistent with the abbreviations in the Allen Brain Reference Atlas.

**Reviewer #2 (Recommendations for the authors):**
(1) What are the criteria for distinguishing large, small, and non-responders?

Large are in the nA range, small are in the hundreds of pA, and non-responders are effectively zero. Manual curation of these responses indicated that a current amplitude threshold of 45 pA clearly separated non-responders from responders. To be clear, the average response (as stated in text and displayed in Figure 3D) includes all cells.

(2) p1. "Unexpectedly": it would not be unexpected, rather, expected, because it was reported in Chen et al., 2023, Nat Neurosci.

The PCG was hinted at, but an actual functional, anatomical connection was not reported in our previous manuscript.

(3) p1. "We combined electrophysiological recordings with immunohistochemistry to assess the molecular identities of these PC targets": please clarify "these" here. It could be read that it refers to "pontine central gray and nearby subnuclei" but it doesn't make sense. Immuno has only been performed for MeV and LC.

Corrected

(4) p1. "but only inhibit a small fraction of cells in many nuclei": as far as I read Fig.3, it seems that ~50% for PBN/VN and ~25% for PCG: would this be "a small fraction"?

The small fraction of cells was in reference to subnuclei within the PCG, but we agree this statement is too broad to be useful and have eliminated it.

(5) p2. "conventional tracer": viral tracer is becoming a standard, so dye tracer could be better here.

Corrected

(6) p3. "rostral/cauda": typo.

Corrected.

(7) p3. Quantification of PC synapses in the brainstem: it would be helpful to introduce why synapto-tdT alone is not sufficient, and the purpose of adding vGAT immunostaining.

We have added more on vGAT labeling putative presynaptic sites and quantifying only synaptic labeling instead of axonal tdTomato in the Results, “Quantification of PC synapses in the brainstem.” In addition, vGAT staining allows us to examine the PC contribution to total inhibition in each region.

(8) p7. "PB and are": typo.

Corrected. And all instances of PBN were changed to PB

(9) p7. "they are likely a mix of excitatory and inhibitory inputs 54,55": Bagnall et al., 2009, J Neurosci, would be critically relevant here.

Added, thank you

(10) Figures 2-3: Yellow/Blue color scheme is hard to distinguish, and having two colors could be read as implying two distinct regions.

We are unsure what the reviewer is referring to exactly here, but the colors refer to the sections in 2C (see the color bar on the bottom right of each atlas schematic). The points represent an individual section that was quantified, and thus do represent distinct samples from distinct regions.

(11) Figure 2D-F: what is indicated by each point?

Each data point is the number of PC bouton (D), density of bouton (E), or percentage of synaptophysin/vGAT (F) quantified for each region per section. Each color represents a coronally distinct section of a region. Additional text was added into the captions to clarify this and point 10.

(12) Figure 3E, right: what is the correlation coefficient?

The correlation coefficient was found to be 0.74

**Reviewer #3 (Recommendations for the authors):**
Some minor grammatical errors and typos need to be cleaned up (e.g. "To quantifying the densities...", "The medial-ventral region of the PBN...have extensive...)".

These errors have been corrected